# Dissecting Post-Training: Uncovering the Complementary Roles of SFT and RL for Document Parsing

Jun-Peng Jiang [1 2 3 *]   An-Yang Ji [1 2 3 *]   Shiyin Lu [3]   Guodong Zheng [4 3 *]   Weihong Zhang [5 3 *]
Qing-Guo Chen [3]   Weihua Luo [3]   Kaifu Zhang [3]   Long Chen [6]   De-Chuan Zhan [1 2]   Han-Jia Ye [1 2]

## Abstract

Document parsing, the task of extracting diverse content from PDFs while preserving their structural integrity, has been significantly advanced by Multimodal Large Language Models (MLLMs). These models have achieved remarkable success, largely driven by extensive post-training on massive datasets. This paper therefore undertakes a deep analysis of the two dominant adaptation strategies, Supervised Fine-Tuning (SFT) and Reinforcement Learning (RL), prompted by a puzzling observation on the PDF-to-Markdown task: SFT makes a negligible impact, especially on parsing complex tables and formulas, while RL achieves substantial overall gains. To unravel the reasons, our systematic investigation reveals a clear and complementary division of labor: SFT primarily operates as a structure learner, biased towards mastering the low-entropy syntax of document layouts. While it learns the format of a table, it struggles to ensure the fidelity of its high-entropy cell content. Conversely, RL excels as a content refiner by optimizing a holistic reward that reflects final accuracy. We further ground this phenomenon in the distinct theoretical nature of their respective objective functions. Based on these findings, we introduce a unified strategy that explicitly harnesses their individual strengths while mitigating their weaknesses. This work shows that a deep understanding of post-training methods is key to unlocking performance beyond what data scaling alone can achieve.

---

[*]Work done during an internship at Alibaba Group [1]School of Artificial Intelligence, Nanjing University [2]National Key Laboratory for Novel Software Technology, Nanjing University [3]Alibaba Group [4]Fudan University [5]Peking University [6]HKUST. Correspondence to: Han-Jia Ye <yehj@lamda.nju.edu.cn>.

*Proceedings of the $43^{rd}$ International Conference on Machine Learning*, Seoul, South Korea. PMLR 306, 2026. Copyright 2026 by the author(s).

## 1. Introduction

Document parsing (Kim et al., 2022; Hwang et al., 2021; Zhang et al., 2024; Ding et al., 2024) is a critical process that transforms unstructured, visually-rich documents like PDFs into structured, machine-readable formats. The core challenge lies in simultaneously preserving the document's structural integrity—such as the layout of tables and the hierarchy of headings—while ensuring the high fidelity of its diverse content, including text and mathematical equations. The recent emergence of powerful Multimodal Large Language Models (MLLMs) (Chen et al., 2024; Bai et al., 2025; Lu et al., 2025) has provided a promising new paradigm for holistically tackling this dual challenge.

Unlike traditional methods (Vik, 2025; Wang et al., 2024b), which rely on rigid pipelines of discrete sub-tasks like layout detection and OCR, MLLMs approach parsing as a unified, end-to-end generation problem. Their remarkable capabilities are typically unlocked through extensive post-training on vast datasets, where the model learns to directly map a document image to a structured output like Markdown (Li et al., 2025; Poznanski et al., 2025; Cui et al., 2025a). This deep reliance on the adaptation phase elevates the choice of post-training strategy from a mere implementation detail to a critical research question, demanding a deeper analysis of its intrinsic behaviors.

Supervised Fine-Tuning (SFT) (Luo et al., 2025; Shi et al., 2024) and Reinforcement Learning (RL) (Bai et al., 2022; Ouyang et al., 2022) are the two dominant post-training strategies for model adaptation. SFT offers a direct learning path by teaching a model to mimic high-quality, expert demonstrations, whereas RL optimizes a model's generation policy through trial-and-error, guided by a reward signal. To test the performance of each strategy, we utilize a large-scale synthetic dataset, meticulously crafted to ensure both data quality and diversity. Intuitively, one might expect SFT to hold a distinct advantage in document parsing. The dense supervision provided by meticulously labeled data should, in principle, offer a stronger and more reliable learning signal than the sparse, scalar rewards of RL, especially from a cold start (Guo et al., 2025).

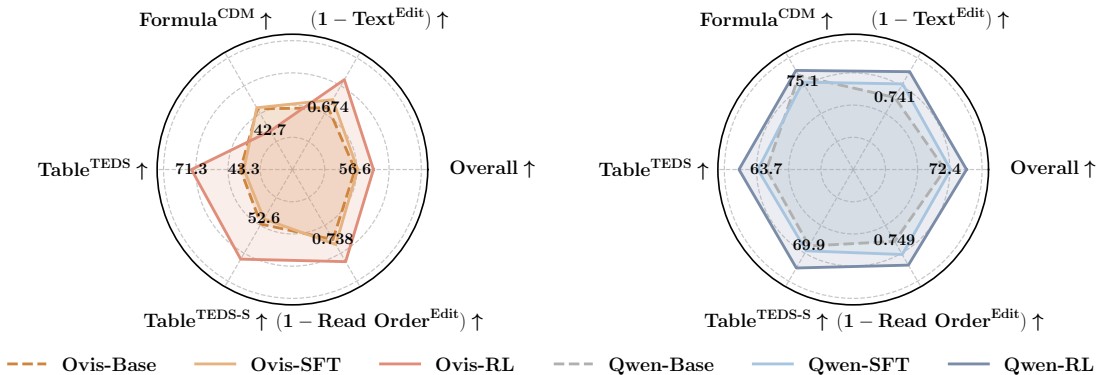

*Figure 1.* Performance of base models and post-training strategies on the OmniDocBench benchmark (Ouyang et al., 2025). Results are shown for Ovis-2B (Lu et al., 2025) (left) and Qwen2.5-VL-2B (Bai et al., 2025) (right) on our synthetic dataset (details in Section 3.2). All metrics are normalized such that a larger area is better, as we show $1 - \text{Text}^{\text{Edit}}$ here. A striking and counter-intuitive pattern emerges: standard Supervised Fine-Tuning (SFT) provides negligible gains over the base model, whereas Reinforcement Learning (RL) yields substantial improvements across nearly all metrics. This puzzling discrepancy forms the central motivation for our investigation into what SFT and RL fundamentally learn.

However, our preliminary experiments directly challenge this intuition. As illustrated in Figure 1, we observe a striking and counter-intuitive outcome on the OmniDocBench benchmark (Ouyang et al., 2025): standard SFT yields negligible performance gains over the base model, demonstrating a particular inability to improve on the complex parsing of tables and mathematical formulas. In stark contrast, RL provides a substantial and consistent improvement across almost all metrics, driven only by simple edit distance as reward. This puzzling discrepancy compels us to ask a fundamental question: *In the context of document parsing, what do SFT and RL actually learn, and why do their behaviors diverge so dramatically?*

To answer this question, we begin by systematically analyzing the factors contributing to the ineffectiveness of SFT. Given that our initial experiments were conducted with high-fidelity synthetic data, we could largely rule out data quality as a primary culprit. This directed our investigation towards other potential factors: the scale of training data, the distribution of content within it, and, most critically, the formulation of the learning task itself. Our empirical analysis revealed that neither simply increasing the data scale nor optimizing the data distribution yielded significant improvements for the SFT process in document parsing. Instead, we found a breakthrough when we reframed the task: instructing the model to generate an intermediate structured representation like JSON, rather than direct Markdown, led to a dramatic and consistent performance boost, as shown in Figure 2.

We further analyze these phenomena from the perspective of their underlying learning objectives. SFT, driven by Maximum Likelihood Estimation, learns a broad distribution to cover all training examples. This biases the model towards mastering the predictable, low-entropy structural syntax over the diverse, high-entropy textual content, making it an effective *structure learner*. In contrast, RL's objective is not to replicate a data distribution but to maximize an expected reward by exploring and learning from its own generated outputs. This allows the learning signal to be tied directly to holistic properties, such as the global edit distance with label, making the model focus on long sequence such as text and HTML tables. This goal-oriented optimization makes RL an inherently effective *content refiner*.

Based on this analysis, we conclude that SFT and RL possess complementary biases. To harness this synergy, we propose a straightforward yet effective two-stage strategy designed to mitigate the inherent limitations of each method. In the SFT stage, we move beyond simple end-to-end training and introduce auxiliary content-focused sub-tasks, compelling the model to learn the intricacies of elements like tables and equations alongside the overall document structure. Subsequently, during the RL stage, we enrich the reward function with explicit structural incentives and fine-grained content-level bonuses. This dual-pronged approach ensures that the model learns to balance both structural integrity and content fidelity throughout the entire post-training process, leading to a more robust document parsing capability. Our contributions are threefold:

- We identify and systematically investigate a counter-intuitive phenomenon in document parsing where standard SFT fails while RL succeeds, providing a clear benchmark of their performance divergence.
- We provide a unified explanation for this phenomenon. Through extensive experiments and a theoretical analysis, we reveal a fundamental division of labor: SFT learns structure, while RL refines content.
- Based on this analysis, we propose a synergistic strategy

to make the best of both worlds in SFT and RL to learn structure and content collaboratively, achieving continuous performance improvement.

# 2. Related Work

## 2.1. Document Parsing

Document Parsing (Kim et al., 2022; Hwang et al., 2021; Zhang et al., 2024; Ding et al., 2024) aims to convert visually-rich, unstructured documents, such as PDFs and images, into structured, machine-readable formats. This process involves not only recognizing textual content but also identifying and interpreting complex layout elements like tables, figures, and hierarchical headings. Historically, this challenge has been tackled by pipeline-based systems that decompose the task into a series of discrete stages, including layout analysis, OCR, and the subsequent reassembly of recognized content, such as MinerU (Wang et al., 2024b), Marker (Vik, 2025), and PP-StructureV3 (Cui et al., 2025b), show great potential for modern document intelligence, this capability underpins a vast spectrum of applications, from advanced document understanding (Appalaraju et al., 2021; Van Landeghem et al., 2023) to intelligent healthcare (Isah & Byström, 2020; Mistry & Arzeno, 2023).

The advent of Multimodal Large Language Models (MLLMs) has ushered in a new era of end-to-end document parsing. Current MLLM-based approaches can be broadly categorized into general-purpose models and specialized document parsing models. General-purpose MLLMs (Team et al., 2023; Chen et al., 2024; OpenAI, 2024; Bai et al., 2025; Lu et al., 2025; Wang et al., 2025b; Xiaojin et al., 2026; Xiangrui et al., 2026) exhibit impressive zero-shot and few-shot parsing capabilities due to their extensive pre-training. Concurrently, a new line of research focuses on developing models specifically tailored for the document domain (Wei et al., 2024; Agrawal et al., 2024; Poznanski et al., 2025; Feng et al., 2025; Ruizhe et al., 2026; Wanli et al., 2026). For instance, MonkeyOCR (Li et al., 2025) leverages a Structure-Recognition-Relation (SRR) triplet paradigm, which simplifies a complex multi-tool pipeline and avoids the inefficiencies of processing full pages with giant end-to-end models. The dots.ocr (RedNote, 2025) is good at multilingual document parsing and unifies layout detection and content recognition within a single vision-language model while maintaining reading order. PaddleOCR-VL (Cui et al., 2025a) integrates a NaViT-style dynamic resolution visual encoder with a language model to enable accurate element recognition, which supports 109 languages and excels in recognizing complex elements. While the power of these models stems from massive-scale pre-training, fully unlocking their potential for specialized tasks depends on the subsequent adaptation phase. Therefore, a fundamental understanding of how different post-training strategies behave is essential for effective and efficient model specialization.

## 2.2. Post-Training in Large Models

Post-training has become the standard paradigm for specializing large pre-trained models to downstream tasks and aligning them with user intent. This landscape is currently dominated by two primary strategies: Supervised Fine-Tuning (SFT) (Luo et al., 2025; Shi et al., 2024) and Reinforcement Learning (RL) (Bai et al., 2022; Ouyang et al., 2022). SFT, or instruction tuning, adapts models via maximum likelihood estimation on curated datasets of expert demonstrations. While effective for endowing models with baseline zero-shot capabilities (Wei et al., 2021; Sanh et al., 2021; Mishra et al., 2022), its reliance on static, off-policy data can lead to overfitting superficial patterns. Conversely, RL adapts the model by optimizing a scalar reward signal, often derived from human preferences (RLHF). The canonical work of Ouyang et al. (2022) demonstrated that RL-aligned models were strongly preferred by humans, cementing its role in alignment. While Proximal Policy Optimization (PPO) (Schulman et al., 2017) remains a standard algorithm, recent critic-less methods like Group Relative Policy Optimization (GRPO) (Shao et al., 2024) have shown particular promise for tasks with verifiable rewards.

Comparative analyses of these paradigms reveal distinct learning dynamics beyond simple performance trade-offs. A primary finding, summarized by Chu et al. (2025), is that SFT tends toward "memorization" of training distributions, whereas RL often yields more robust, generalizable policies. Furthermore, contrasting conventional intuition, recent studies (Chen et al., 2025; Shenfeld et al., 2025; Lai et al., 2025) indicate RL-tuned models frequently exhibit less catastrophic forgetting than their SFT counterparts. This robustness is often attributed to "RL's Razor" (Shenfeld et al., 2025), arguing that on-policy methods implicitly bias updates to remain closer to the base model's knowledge distribution. Existing literature largely frames these methods as competitors. However, their distinct inductive biases may be synergistic for complex, composite tasks like document parsing, which demands simultaneous mastery of low-entropy structural syntax (e.g., Markdown) and high-entropy semantic content. We investigate this intersection, positing that SFT and RL are complementary in this domain: the former serves as an effective structure learner, while the latter acts as a proficient content refiner.

# 3. Preliminaries

In this section, we first formalize the task of post-training for document parsing and establish some notations. We then outline the experimental setup used for our empirical comparison of SFT and RL.

### 3.1. Post-Training for Document Parsing

We formulate the document parsing task as a conditional generation problem. Let $D$ represent a document image. The objective is to generate a corresponding structured text sequence $Y = (y_1, y_2, ..., y_m)$. The sequence $Y$ is a high-fidelity representation that preserves both the content and the reading order of the document $D$ according to the overall structure. The document may contain various semantic elements, including multi-level headings, paragraphs, figures, tables, and formulas. For complex elements, $Y$ employs specific markup languages to maintain structural integrity. For instance, tables are converted into HTML format, mathematical equations are represented in LaTeX, and other textual content is transcribed as plain text.

Current modern approaches adapt a pre-trained Multimodal Large Language Model (MLLM) with post-training methods in a large amount of data, which we model as a policy $\pi_\theta(Y|D)$ that generates a sequence $Y$ conditioned on the document $D$. The goal of post-training is to adapt the initial model parameters $\theta_{\text{pre}}$ to a new set $\theta$ specialized for this task. We investigate two primary post-training strategies:

**Supervised Fine-Tuning (SFT)** aims to minimize the cross-entropy loss with respect to ground-truth sequences $Y^*$. Given a dataset of expert demonstrations $(D, Y^*)$ in the ground truth distribution $\mathcal{D}$, the objective is:

$$\mathcal{L}_{\text{SFT}}(\theta) = -\mathbb{E}_{(D,Y^*)\sim\mathcal{D}}\left[\log \pi_\theta\left(Y^* \mid D\right)\right]. \quad (1)$$

**Reinforcement Learning (RL)**, in contrast, optimizes the policy by maximizing a reward signal from a function $r(D, Y)$ that evaluates the quality of the generated sequence. The objective, typically regularized by the KL-divergence from the initial policy $\pi_{\theta_{\text{pre}}}$ to maintain stability, is formulated as:

$$\mathcal{J}_{\text{RL}}(\theta) = \mathbb{E}_{D\sim\mathcal{D}, Y\sim\pi_\theta(\cdot|D)}[r(D, Y)] - \\ \beta \cdot \text{KL}\left[\pi_\theta(\cdot \mid D)\|\pi_{\theta_{\text{pre}}}(\cdot \mid D)\right], \quad (2)$$

where $\beta > 0$ is the regularization coefficient.

### 3.2. Preliminary Setup

**Datasets.** To ensure a controlled and fair comparison while guaranteeing high data quality, we developed a synthetic data generation pipeline. This approach allows us to create perfectly aligned image-Markdown pairs, eliminating the noise and ambiguity often present in real-world scanned documents.

Our pipeline begins with the procedural generation of Markdown content. We construct documents with a varied hierarchical structure by creating multi-level headings populated with words and phrases randomly sampled from a large text corpus (Schuhmann et al., 2022; Changpinyo et al.,

2021). To introduce complex, non-textual elements, we integrate tables and mathematical equations randomly sampled from the established PubTabNet (Smock et al., 2022) and UniMerNet (Wang et al., 2024a) datasets. These elements are seamlessly embedded within the Markdown source. The complete Markdown document is then rendered into an HTML page. To simulate the diversity of real-world document layouts, we apply a wide range of CSS stylesheets to the HTML, varying fonts, spacing, column layouts, and other visual properties. Finally, each styled HTML page is rendered and saved as a random resolution document image. This process yields a dataset of 100k image-Markdown pairs, providing a robust and clean foundation for our experiments. A more detailed data information can be found in Appendix.

**Models and Baselines.** To validate our findings, we conduct experiments on two recent and powerful general MLLMs: Qwen2.5-VL (Bai et al., 2025) and Ovis2.5 (Lu et al., 2025). Recognizing that many state-of-the-art document parsing models are specialized and often smaller in scale, we intentionally select the 2-billion parameter variants of these models, specifically Qwen2.5-VL-3B and Ovis2.5-2B.

Our primary comparison is between the two main post-training strategies SFT and RL. For SFT, the model is fine-tuned on the synthetic dataset using the standard cross-entropy loss. For RL, we employ Group Relative Policy Optimization (GRPO) (Guo et al., 2025) to optimize the policy. For the reward function $r(D, Y)$, we use the normalized edit distance between the generated sequence $Y$ and the ground-truth $Y^*$. The reward is formulated as $1 - \text{NED}(Y, Y^*)$, where NED is the Levenshtein distance normalized by the length of the longer sequence.

**Evaluation Metrics.** We evaluate our models on OmniDocBench (Ouyang et al., 2025), a comprehensive benchmark for assessing document parsing in real-world scenarios. The accuracy of plain text is evaluated using Edit Distance to quantify character-level similarity. For tables, which are structured as HTML, we report the Tree Edit Distance-based Similarity (TEDS), as well as TEDS-Structure (TEDS-S) (Zhong et al., 2020). The fidelity of mathematical equations, represented in LaTeX, is assessed with the Character Detection Matching (CDM) (Wang et al., 2025a), which calculates the pixel-level similarity between the rendered images of the generated and ground-truth formulas.

Our evaluation is designed to quantify the impact of each post-training strategy. Therefore, for any given metric $M$, we focus on reporting the performance gain, denoted as $\delta_M$, relative to the initial, pre-trained base model. Formally, let $M_{base}$ be the score achieved by the base policy $\pi_{\theta_{\text{pre}}}$ and $M_{post}$ be the score of the post-trained policy $\pi_\theta$. The gain or drop is then defined as $\Delta M = M_{post} - M_{base}$. A higher $\Delta M$ signifies a more effective adaptation process.

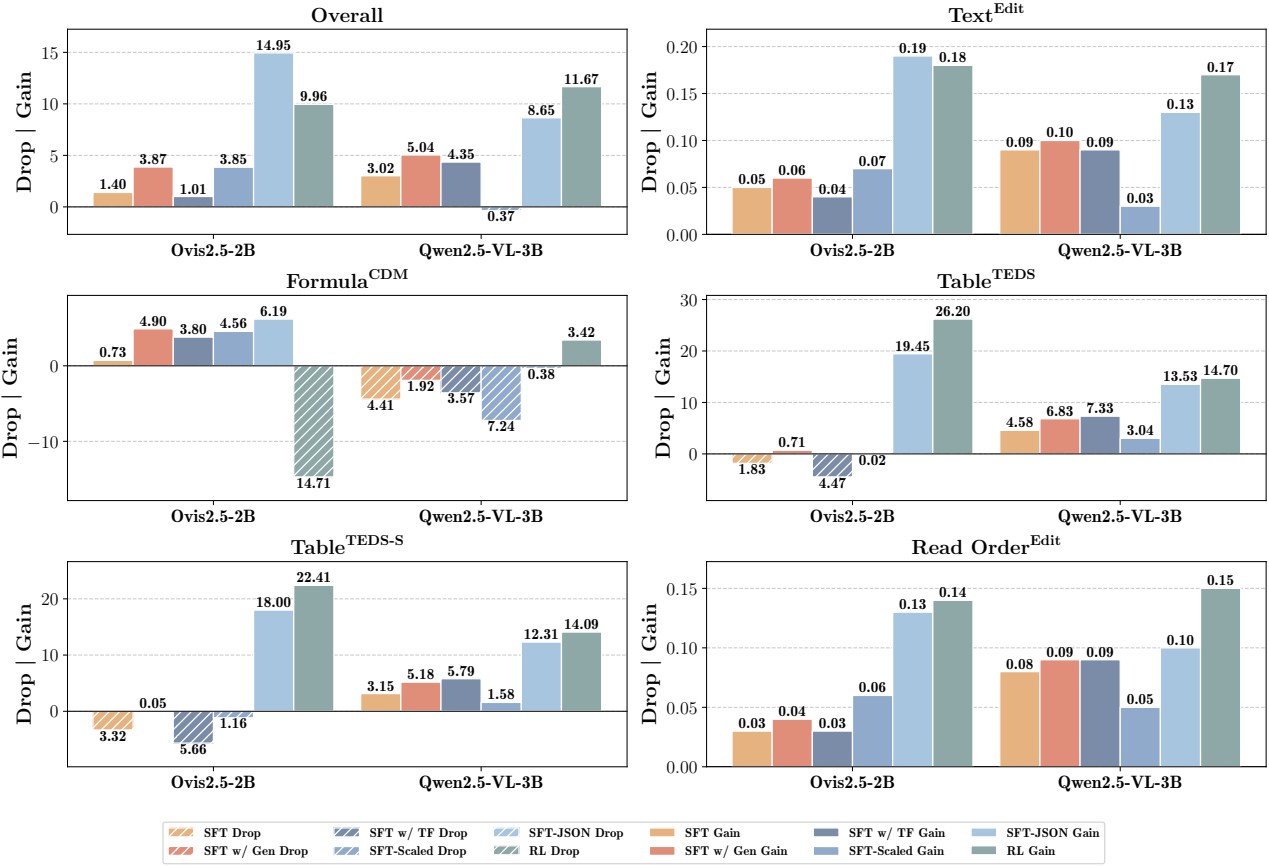

*Figure 2.* Systematic analysis of SFT's ineffectiveness. We investigate several hypotheses for SFT's failure by augmenting the training data with general-purpose data (SFT w/ Gen) , targeted table/formula data (SFT w/ TF) , and by tripling the training data size (SFT-Scaled). The results show that none of these data-centric strategies yield significant performance gains. In contrast, reframing the learning goal from direct Markdown generation to predicting a structured JSON (SFT-JSON) leads to a dramatic performance surge across both models, achieving gains nearly on par with the powerful RL baseline. This provides strong evidence that SFT's failure is not due to the data but is fundamentally linked to the learning task's formulation.

## 4. What do SFT and RL Learn in Document Parsing

In this section, we analyze the factors behind the divergent performance of SFT and RL. We first present a series of empirical studies exploring their differing sensitivities to data and task formulation, and then connect these findings to the fundamental nature of their learning objectives.

### 4.1. Empirical Analysis for Behavioral Differences

We begin our analysis by investigating the root causes of SFT's ineffectiveness, as initially observed and depicted in Figure 2. We systematically test three plausible hypotheses for this failure: a loss of generalist capabilities, insufficient complex structures, and an inadequate scale of training data.

**Learning with General Data.** Our first hypothesis was that fine-tuning on the specialized document parsing task might cause the model to forget its more general vision-language abilities, a phenomenon known as catastrophic forgetting (Ramasesh et al., 2021; Zhai et al., 2023; Luo et al., 2025). To test this, we augmented our 100k synthetic dataset with 50k samples from diverse, general-purpose tasks, including image captioning, visual question answering (VQA), and general scene understanding. As shown in Figure 2, this data mixture did not yield any significant performance improvements, suggesting that a loss of general capabilities is not the primary issue.

**Learning with Targeted Data.** Next, we hypothesized that the model's poor performance, particularly on table and formula metrics, was due to a lack of targeted exposure to these complex structures. We addressed this by supplementing the training data with an additional 10k specialized samples, focusing exclusively on table- and formula-rich documents. Counter-intuitively, this targeted training regime also failed to produce a meaningful uplift in the corresponding TEDS and CDM scores. The results indicate that the model struggles to acquire these specific skills through SFT, even with

dedicated data.

**Learning with Scaled Data.** Finally, we explored the possibility that the issue was simply a matter of scale. We tripled our training dataset from 100k to 300k samples, following the same generation process. While this brute-force scaling did result in marginal gains across most metrics, the improvements were trivial compared to the substantial increase in data and computational cost. This finding strongly suggests that the limitations of SFT in this context are not merely a function of data quantity but are indicative of a more fundamental issue with the learning approach itself.

**Learning with Another Format.** The failures of data-centric solutions led us to a new hypothesis: the limitation might not lie in the data itself, but in the formulation of the learning task. The direct end-to-end generation of a complex Markdown sequence forces the model to simultaneously learn layout semantics, content transcription, and syntactic formatting. Inspired by traditional pipeline approaches that decouple layout detection from content recognition, we conjectured that simplifying the learning objective could unlock the model's capabilities.

To validate this, we reframed the parsing task. Instead of generating a raw Markdown string, we instructed the model to output a structured JSON object. In this formulation, the model's primary responsibilities are to (1) identify the semantic type of each document element (e.g., heading, paragraph, table, formula) and (2) transcribe the content of each element in the correct reading order. The burden of generating precise Markdown syntax is thus removed.

The results of this experiment were striking. As shown in Figure 2, models trained on this JSON-based task achieved a performance level nearly on par with the RL-trained models, representing a dramatic improvement over the direct Markdown generation baseline. This finding provides strong evidence that the SFT process is highly sensitive to the task's objective. It excels at learning to classify and structure semantic content when the task is explicitly defined this way, but struggles when this objective is entangled with the complexities of generative syntax.

Summarizing these empirical findings, a clear pattern emerges. Even when the direct Markdown generation failed on final metrics, qualitative analysis of the outputs revealed that the SFT models had successfully learned the overall document layout and the coarse-grained content of each segment. The subsequent success of the JSON task suggests that *SFT's primary achievement was always the acquisition of structural knowledge*; performance only surged when the task was reframed to reward this strength directly. Conversely, the RL-trained models, guided by a holistic reward, demonstrated superior performance on metrics sensitive to long-range accuracy, such as the edit distance of lengthy paragraphs and the TEDS of large tables. However, this global focus sometimes came at the cost of fine-grained elements like short mathematical formulas. This indicates that *RL's strength lies in optimizing for overall content* fidelity, even if it occasionally overlooks local details.

## 4.2. Theoretical Perspective: Structure vs. Content

Our empirical findings can be explained through a deeper analysis of the SFT and RL objectives, viewed through the lens of KL-divergence. We argue that SFT's objective approximates a forward KL minimization, leading to "mode-covering" behavior that favors structure, while RL's objective approximates a reverse KL minimization, promoting "mode-seeking" behavior that refines content.

**SFT as Forward KL: Covering the Structural Modes.** The SFT objective, which minimizes the negative log-likelihood with respect to the ground-truth distribution $P^*(Y|D)$, is equivalent to minimizing the forward KL-divergence $KL(P^*(Y|D)||\pi_\theta(Y|D))$. This formulation is also known to be mode-covering (Chen et al., 2025). In the context of document parsing, the ground-truth distribution $P^*$ is multi-modal; there are many acceptable ways to transcribe content (high variance), but very few correct ways to format the structure (low variance). To cover all modes of $P^*$ and avoid assigning low probability to any ground-truth sequence, the policy $\pi_\theta$ is encouraged to become a broad, average representation of the data.

This averaging behavior is particularly pronounced for the highly-consistent, low-entropy structural elements. The model quickly learns to assign high probability to the common syntactic "modes" (e.g., the exact "#" for headings, "`<table>`" for tables, and "$$" for formulas in Markdown) because they are shared across the entire dataset. Conversely, it struggles with the high-entropy content, as averaging over the vast space of possible text results in a diffuse, non-committal distribution. Therefore, the mode-covering nature of SFT makes it an effective structure learner.

**RL as Reverse KL: Seeking High-Reward Content Modes.** The KL-regularized RL objective can be shown to be equivalent to minimizing the reverse KL-divergence $KL(\pi_\theta(Y|D)||P_r(Y|D))$, where $P_r$ is an implicit target distribution defined by the reward function: $P_r(Y|D) \propto \pi_{\theta_{\text{pre}}}(Y|D)\exp(r(D,Y)/\beta)$ (Korbak et al., 2022). This formulation is mode-seeking (Chen et al., 2025).

Unlike SFT, the policy $\pi_\theta$ is not required to cover the entire space of acceptable outputs. Instead, it is encouraged to find and concentrate its probability mass on the high-reward modes within the distribution $P_r$. In our task, the reward $r(D,Y)$ is highest for sequences with perfect content fidelity. The RL process will therefore actively seek out and exploit policies that generate the correct text, even if those

policies represent a very narrow "mode" in the vast space of possible sequences. The model learns to be a specialist, focusing its probability mass on the generation paths that lead to high content scores. This mode-seeking behavior makes RL an extremely effective content refiner.

**Dataset-Level Evidence of Mode-Covering and Mode-Seeking.** The above KL-based interpretation is also reflected in the macro-level quantitative patterns observed across the full evaluation set. For SFT, minimizing the negative log-likelihood can be viewed as minimizing the forward KL divergence,

$$
\mathrm{KL}(P^\star(Y|D) \,\|\, \pi_\theta(Y|D)) = \mathbb{E}_{Y \sim P^\star} \left[ \log \frac{P^\star(Y|D)}{\pi_\theta(Y|D)} \right],
\tag{3}
$$

which strongly penalizes the model when it assigns very low probability to valid regions of the ground-truth distribution. This mode-covering property naturally encourages the model to cover the common structural modes shared by the dataset. Empirically, this explains why SFT is particularly effective at learning low-entropy layout rules. For example, the severe format failure rate is reduced from approximately $50.4\%$ for the base model to nearly zero after SFT, indicating that the model successfully captures common structural constraints such as the JSON schema and document layout patterns. However, because forward KL encourages probability mass to be spread over the support of the ground-truth distribution, it may lack the sharpness required for highly specific and high-entropy content generation. This is consistent with our observation that SFT quickly plateaus on content-sensitive metrics, such as formula CDM, even when more training data is provided.

In contrast, RL with reward maximization is closely related to a reverse-KL-like objective,

$$
\mathrm{KL}(\pi_\theta(Y|D) \,\|\, P_r(Y|D)) = \mathbb{E}_{Y \sim \pi_\theta} \left[ \log \frac{\pi_\theta(Y|D)}{P_r(Y|D)} \right],
\tag{4}
$$

where $P_r(Y|D)$ denotes the implicit reward-induced target distribution. This mode-seeking behavior encourages the policy to concentrate its probability mass on high-reward outputs while suppressing low-reward generations. In document parsing, such a property is well aligned with content refinement, since the reward assigns higher values to outputs with more accurate transcriptions. This explains why pure RL can substantially improve character-level and content-sensitive metrics, such as edit distance and CDM. Nevertheless, because the reward-dominant modes are mainly determined by content fidelity, pure RL may under-emphasize structural constraints that are not sufficiently captured by the reward. This is reflected by the fact that pure RL can underperform SFT-JSON on structure-sensitive metrics such as TEDS for complex tables. Moreover, when pure RL is

directly applied to fragile structured outputs such as JSON, invalid generations may receive zero reward, resulting in unstable optimization or even training collapse.

Overall, the divergence between SFT and RL in our quantitative results is therefore not an isolated empirical artifact. Rather, it is a dataset-level manifestation of their different optimization biases: SFT behaves as a mode-covering structure learner that reliably captures low-entropy formatting and layout rules, whereas RL behaves as a mode-seeking content refiner that pushes the model toward high-reward, high-fidelity transcriptions.

## 5. Make the Best of Both Worlds

Our analysis has revealed that SFT and RL are not competing but complementary paradigms, each with a distinct inductive bias: SFT excels at learning structure, while RL is proficient at refining content. A naive application of either method in isolation, or a simple sequential pipeline, fails to leverage their full potential. This section introduces a unified strategy designed to explicitly harness their individual strengths and mitigate their weaknesses, ensuring that both structure and content are learned robustly throughout the post-training process.

### 5.1. Content-Enriched SFT for Structural Scaffolding

The primary limitation of standard SFT is its tendency to prioritize low-entropy structural patterns over high-entropy content. To counteract this, we propose a Content-Enriched SFT stage. The goal is not only to learn the document's layout but also to build a strong baseline understanding of its content. We achieve this by augmenting the primary JSON-based parsing task with an auxiliary content objective.

Specifically, alongside the main task of predicting the document's JSON structure, we introduce a set of content-intensive sub-tasks. These include:

- Formula Recognition: For document regions identified as formulas, we task the model with transcribing the corre-

*Table 1.* Ablation study of our Content-Enriched SFT. Adding specialized data for tables and formulas (w/ TF) to the JSON-based SFT task consistently improves performance on content-sensitive metrics across both models. This demonstrates that targeted content enrichment successfully mitigates SFT's bias against content learning. Metrics are defined as follows: OA (Overall), ED (Edit Distance for text) , and Read (Read Order Edit).

|  | OA | ED | CDM | TEDS | TEDSS | Read |
|---|---|---|---|---|---|---|
| Ovis | 71.58 | 0.13 | 63.59 | 64.55 | 73.91 | 0.13 |
| w/ TF | 82.43 | 0.09 | 80.50 | 75.59 | 81.29 | 0.09 |
| Qwen | 79.41 | 0.15 | 77.87 | 74.98 | 79.80 | 0.17 |
| w/ TF | 81.09 | 0.13 | 79.16 | 77.20 | 82.19 | 0.15 |

*Table 2.* Ablation study of our Structure-Aware RL. We analyze the impact of our decomposed reward function (Ours) versus a standard global reward (Base), both with and without extra table/formula data (w/ TF). (1) Our decomposed reward consistently outperforms the base reward, significantly boosting TEDS and CDM scores by providing targeted learning signals. (2) Adding specialized TF data further improves content-sensitive metrics. This shows that while better data helps, the structure-aware reward is the critical factor in balancing the model's performance across all document elements.

| | | OA ↑ | ED ↓ | CDM ↑ | TEDS ↑ | TEDSS ↑ | Read ↓ |
|---|---|---|---|---|---|---|---|
| Ovis2.5-2B | RL (Base) | 66.59 | 0.14 | 42.69 | 71.3 | 78.32 | 0.12 |
| | RL (Decomposed) | 71.62 | 0.18 | 72.20 | 60.97 | 69.03 | 0.16 |
| | RL w/ TF (Base) | 78.04 | 0.17 | 83.62 | 67.03 | 72.30 | 0.15 |
| | RL w/ TF (Decomposed) | 79.00 | 0.17 | 85.30 | 69.01 | 74.98 | 0.17 |
| Qwen2.5-VL-3B | RL (Base) | 84.11 | 0.09 | 82.97 | 78.38 | 83.97 | 0.10 |
| | RL (Decomposed) | 84.29 | 0.09 | 84.35 | 77.64 | 82.92 | 0.10 |
| | RL w/ TF (Base) | 84.64 | 0.09 | 84.94 | 78.18 | 84.12 | 0.10 |
| | RL w/ TF (Decomposed) | 85.25 | 0.09 | 86.44 | 78.63 | 83.44 | 0.11 |

sponding LaTeX string.

- Table Transcription: For tables, the model is trained to accurately extract the HTML from each individual cell.

These auxiliary tasks are co-trained with the main structural parsing objective. By forcing the model to pay explicit attention to the high-fidelity details of content during the SFT phase, we prevent it from solely optimizing for structural mimicry. This produces a model that is not only a competent "architect" of the document's layout but also a capable "librarian" of its content, providing a much stronger foundation for the subsequent RL stage.

As shown in Table 1, the efficacy of our content-enrichment strategy is clear. The introduction of specialized table and formula data during the SFT stage provides a distinct boost to the TEDS and CDM metrics, respectively. This improvement holds consistently across both the Ovis and Qwen models, confirming that these targeted sub-tasks successfully enhance the model's content recognition capabilities beyond learned from the general parsing task alone.

### 5.2. Structure-Aware RL for Holistic Refinement

The standard RL approach, while effective for content, can suffer from "reward hacking" or a myopic focus. A global reward like edit distance might disproportionately favor improvements in long paragraphs while neglecting smaller but equally critical elements like single-line equations or list items. To address this, we introduce a Structure-Aware RL stage, utilizing a decomposed reward function.

Our reward $r(D, Y)$ is a weighted combination of multiple components, designed to provide a balanced learning signal across all aspects of the document:

$$r(D, Y) = \sum_{i \in \text{ elements}} w_i \cdot r_i, \qquad (5)$$

where $r_i$ indicates fine-grained, element-specific rewards. In our paper, they are $r_{\text{text}}$ for text Edit Distance, $r_{\text{table}}$ for

TEDS and TEDS-S, and $r_{\text{formula}}$ for formula CDM. $w_i$ is the weight to balance the different reward components. We set all weights the same.

This decomposed reward ensures the model is incentivized to improve every part of the document. A small but perfect correction to a mathematical formula will now receive a direct and significant reward signal, preventing it from being overshadowed by minor tweaks in a long paragraph. This approach compels the reward function to refine the document holistically, paying equal attention to both the overarching content and the correctness of each structural component. By making structure-aware, we ensure it polishes the entire artifact, not just the largest or easiest parts.

The ablation study in Table 2 isolates the impact of our Structure-Aware RL. When trained with a standard, global reward, the models learn to prioritize improvements on general text, showing only modest gains in short CDM metrics. However, the introduction of our decomposed reward function produces a significant shift. Across both models, the structure-aware reward leads to substantial improvements in text, table, and formula scores. This confirms that our method successfully encourages the model to refine the document holistically, ensuring that the content fidelity does not come at the expense of neglecting structural components.

### 5.3. Make the Best of Both Worlds

Having improved each stage individually, we now present the combined performance of our full, unified pipeline. In this approach, the model produced by Content-Enriched SFT serves as the initial policy for our Structure-Aware RL.

Figure 3 contrasts the final performance of this synergistic SFT-RL approach against the base model. The results demonstrate a dramatic leap in capability. While our earlier analysis showed that standard SFT offered negligible improvements and standard RL provided only moderate gains, our unified method unlocks a new level of perfor-

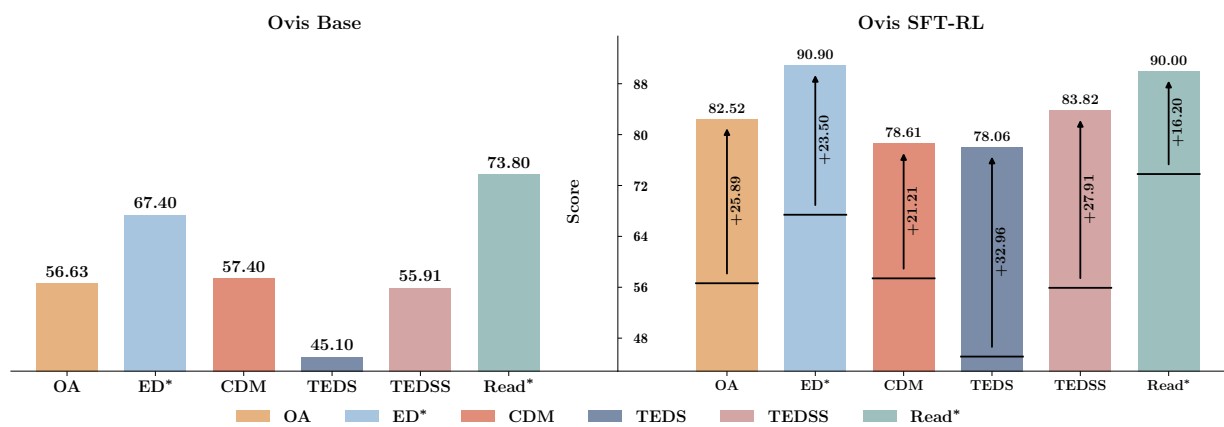

*Figure 3.* Performance comparison of our full, unified SFT-RL pipeline (Ovis SFT-RL) against the Ovis base model (Ovis Base). The unified approach, which initializes Structure-Aware RL with the output of Content-Enriched SFT, achieves substantial gains across all evaluation metrics. This highlights the synergistic effect of a strategy designed to leverage the complementary strengths of SFT (as a structure learner) and RL (as a content refiner). To ensure all metrics are aligned for visualization (where higher is better), the two metrics marked with an asterisk (ED* and Read*) are transformed using the formula: $(1 - \text{Metric}) \times 100$.

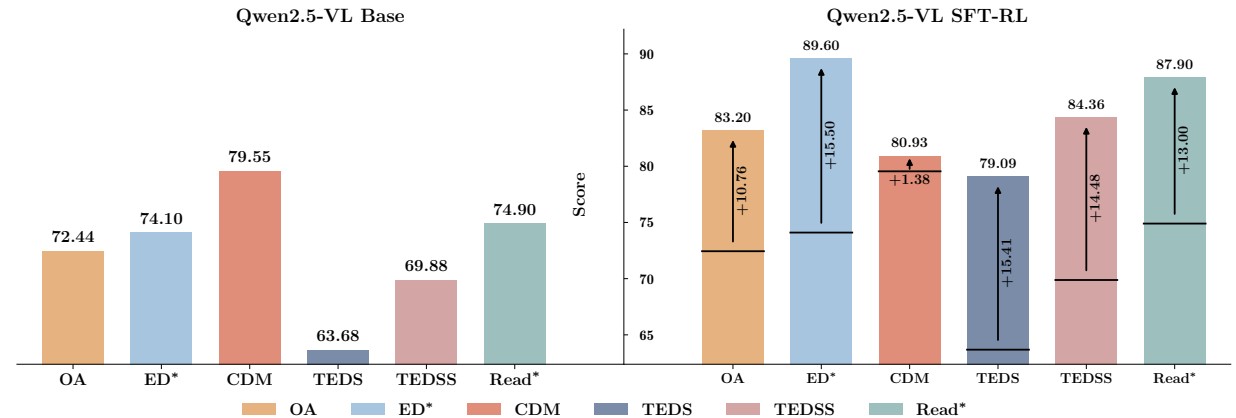

*Figure 4.* Performance comparison of our full, unified SFT-RL pipeline (Qwen2.5-VL SFT-RL) against the Qwen2.5-VL base model.

mance across all metrics. This substantial improvement validates our core thesis. By thoughtfully addressing the inherent biases of each post-training phase—ensuring SFT learns content alongside structure and that RL refines both holistically—we can fully harness their complementary nature. The final result is not merely an incremental improvement but a transformative one, showcasing the power of a principled, insight-driven approach to model adaptation.

For a comprehensive evaluation on the Qwen2.5-VL model for both strategies, which shows consistent improvements in Figure 4. Please refer to the Appendix B for more details and more experiments.

## 6. Conclusion

In this paper, we presented a deep investigation into the contrasting behaviors of Supervised Fine-Tuning (SFT) and Reinforcement Learning (RL) for document parsing. Our

inquiry was motivated by a paradoxical phenomenon: the surprising failure of SFT in a context where RL, against intuition, proved highly effective. This served as the starting point for an analysis that looked beyond surface-level performance to uncover the fundamental mechanics of what each post-training paradigm learns. Our central contribution is the elucidation of a fundamental division of labor: through systematic empirical and theoretical analysis, we demonstrate that SFT is inherently biased towards learning low-entropy structure, while RL functions as a proficient refiner of high-entropy content. Building on these, we demonstrated that a synergistic strategy—one that enriches SFT with content-aware objectives and equips RL with structural incentives—can leverage these complementary strengths to achieve superior performance. We hope our analysis of how post-training methods learn makes insights for moving beyond a brute-force, data-centric view and toward the principled design of truly effective and robust MLLMs.

## Acknowledgements

This work is partially supported by National Key R&D Program of China (2024YFE0202800), Basic Research Program of Jiangsu under Grants (BK20253021), NSFC (62522605, 62376118, 62522216, 62402408), Hong Kong SAR Research Grants Council (RGC) Early Career Scheme (26208924), Hong Kong SAR Research Grants Council (RGC) General Research Fund (16219025), the Fundamental and Interdisciplinary Disciplines Breakthrough Plan of the Ministry of Education of China (No. JYB2025XDXM118), the "111 Center" (No. B26023), the Collaborative Innovation Center of Novel Software Technology and Industrialization.

## Impact Statement

This paper presents work whose goal is to advance the field of Machine Learning, especially in Document Parsing. There are many potential societal consequences in our work related to our model and benchmark, none of which we feel must be specifically highlighted here.

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

# A. Training Details

In this section, we provide a comprehensive breakdown of the data construction process, statistical distributions of our training sets, and the specific implementation details.

## A.1. Dataset Details and Distributions

### A.1.1. SYNTHETIC DATA GENERATION PIPELINE.

To overcome the scarcity of high-quality, pixel-aligned document-markdown pairs, we developed a robust synthetic data generation engine. This engine is designed to simulate real-world PDF production while ensuring noise-free ground truth labels. The pipeline consists of three core modules:

**Content Generation Mechanism:** We utilize the `Faker` library to construct randomized textual elements, including hierarchical headers, paragraphs, and lists, simulating the structural diversity of real documents. To support scientific and technical domains, the engine parses LaTeX equations to generate mathematical content. Additionally, an image integration module dynamically inserts figures from local or networked sources, ensuring visual richness.

**Document Rendering & Conversion:** The core format is Markdown, which ensures structural integrity. We employ `WeasyPrint` to render the Markdown into HTML and subsequently into PDF format. Finally, `pdf2image` is used to rasterize PDF pages into JPG images. This "Markdown-to-PDF-to-Image" workflow guarantees that the visual layout perfectly aligns with the structured ground truth.

**High-Efficiency Batch Processing:** To meet the demands of large-scale pre-training, the engine implements concurrency via Python's `concurrent.futures`, enabling the rapid generation of hundreds of thousands of diverse document samples.

### A.1.2. DATASET DETAILS.

Based on the pipeline described above, we constructed several specific datasets to investigate the behaviors of Supervised Fine-Tuning (SFT) and Reinforcement Learning (RL). The statistics for each dataset are summarized in Table 3.

**Base Dataset.** This is the foundational dataset used for our primary comparisons. It is composed of two main parts: OCR-Centric Data, constructed from non-semantic random words to enforce visual reliance, and Semantic Corpus Data, constructed by stitching together randomly truncated semantic segments to maintain language modeling capabilities.

**Ablation Datasets.** To analyze the limitations of standard SFT, we constructed three variants. For SFT w/ Gen, we augmented the Base dataset with general vision-language instruction data (e.g., image captioning) to test for catastrophic forgetting. For SFT w/ TF, we added specialized pages containing complex tables and formulas to test targeted exposure. For SFT-Scaled, we upscaled the sampling of both OCR and semantic components and included the full set of complex elements to test the impact of data scaling.

**Format-Specific Datasets.** To evaluate the impact of task formulation, we converted the semantic corpus and table/formula subsets into a structured JSON format. This shifts the objective from raw Markdown generation to key-value structured prediction. We provide the specific prompts used to instruct the model for both Markdown and JSON generation in Table 4, demonstrating how the task definition is altered. Furthermore, a visualization of the ground-truth JSON data structure is

*Table 3.* Dataset Statistics and Composition. TF denotes specialized Table and Formula data.

| Dataset Variant | Composition | Total Samples |
|---|---|---|
| **SFT Base** | OCR-Centric + Semantic Corpus | 68924 |
| **SFT w/ Gen** | Base + General Data | 118924 |
| **SFT w/ TF** | Base + Synthetic TF Data | 78821 |
| **SFT-Scaled** | OCR-Centric + Corpus + TF (Upscaled) | 157642 |
| **SFT-JSON** | Semantic Corpus (JSON) + TF (JSON) | 49036 |
| **RL Base** | OCR-Centric + Semantic Corpus | 16842 |
| **RL w/ TF** | Base + Synthetic TF Data | 19827 |

presented in Figure 5, highlighting how structural hierarchy (e.g., headers, paragraphs, tables) is explicitly encoded.

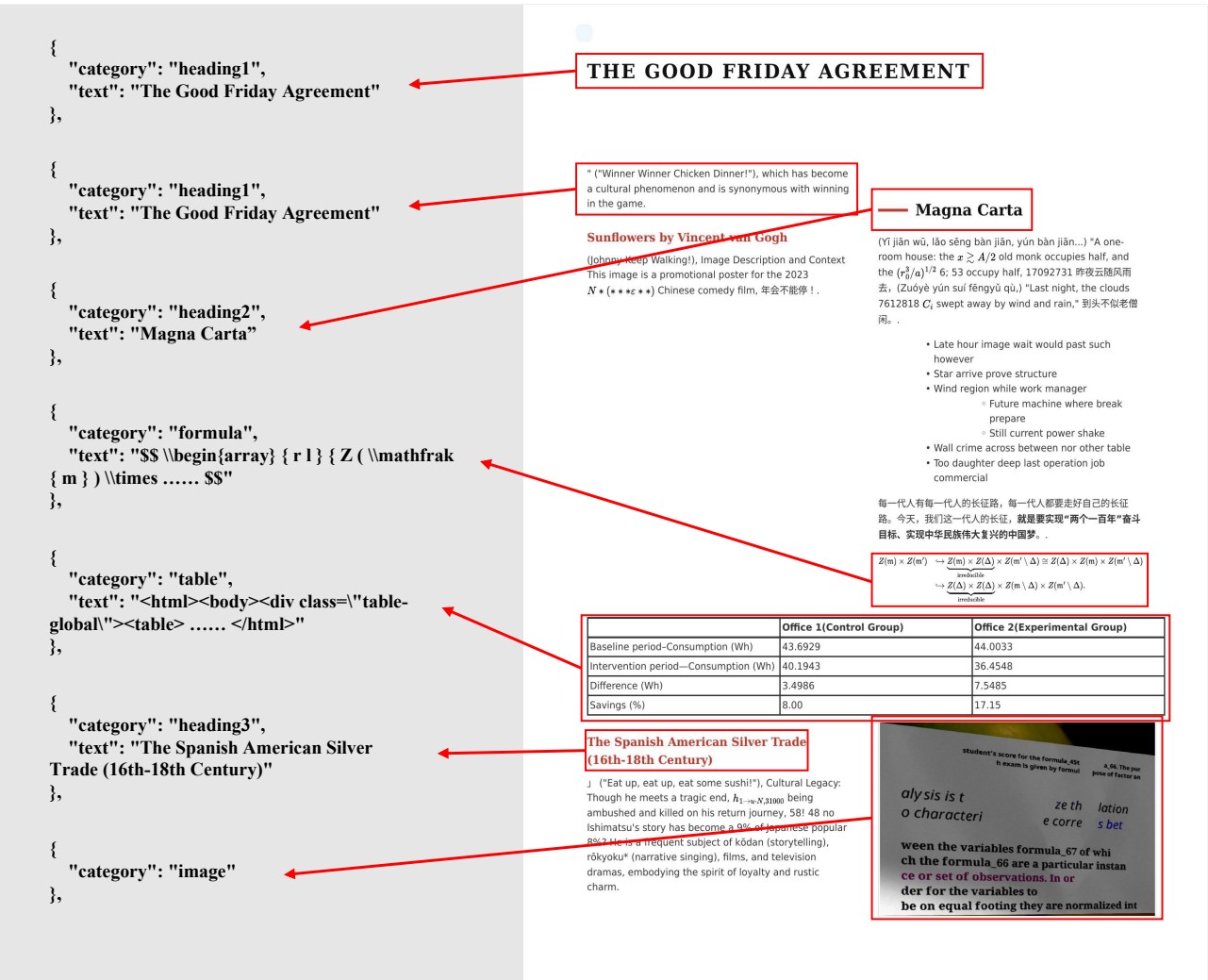

*Figure 5.* Visualization of a sample from the SFT-JSON Dataset. Unlike the raw Markdown sequence, this format forces the model to explicitly categorize semantic regions (e.g., "paragraph", "formula") before generating the content, decoupling structural layout prediction from content transcription.

**RL Datasets.** The RL stage utilizes the same data sources as the corresponding SFT experiments (RL Base and RL w/ TF) to ensure a fair comparison of optimization methods.

### A.1.3. CONTROLLED SETTING AND SCOPE.

The primary goal of this work is to investigate the fundamental behavioral divergence between SFT and RL in document parsing. To make this comparison rigorous, our experimental design prioritizes controlled variables. Real-world document parsing datasets often contain annotation noise, OCR misalignment, inconsistent formatting, and ambiguous ground-truth structures. If such noisy data were used as the main training source, it would be difficult to determine whether a parsing failure is caused by the learning algorithm itself or by imperfections in the supervision. In contrast, our synthetic pipeline provides strictly aligned image-text pairs in a noise-controlled environment, while still introducing visual diversity through varied document layouts and CSS styles. This design allows us to more directly isolate the optimization behaviors of different post-training strategies.

We also focus on 2B–3B scale MLLMs, which represent a practical model scale for high-resolution document parsing, where computational cost grows rapidly with both image resolution and model size. More importantly, our analysis is

not intended to attribute the observed SFT–RL divergence to a particular parameter scale. The KL-based interpretation characterizes the optimization bias of the learning objectives: SFT exhibits a mode-covering tendency, while RL exhibits a mode-seeking tendency. These objective-level properties are not specific to a particular model size, although the empirical magnitude of their effects may vary across architectures and scales.

Finally, we clarify that our conclusions are primarily drawn within the document parsing domain. This domain is especially suitable for studying the complementary roles of SFT and RL because it tightly couples rigid, low-entropy structural constraints with high-entropy content transcription. Therefore, rather than aiming to make a universal claim across all tasks, our study uses document parsing as a representative setting where the division of labor between structure learning and content refinement can be clearly observed and systematically analyzed.

### A.2. Hyperparameters

The training process is divided into two distinct stages: SFT and RL (Shao et al., 2024). For the SFT stage, we train for 1 epoch using a cosine learning rate schedule. For the RL stage, we employ the Group Relative Policy Optimization (GRPO) algorithm. We generate 8 candidate outputs per prompt ($G = 8$) to estimate the baseline. A constant learning rate is used to ensure stability during the reinforcement phase. The detailed hyperparameters for both Ovis2.5-2B and Qwen2.5-VL-3B are provided in Table 5.

## B. More Experiments

### B.1. Detailed Analysis of Ovis2.5 and Qwen2.5-VL

In the main paper, we primarily visualized the complete pipeline results for Ovis2.5 due to space constraints. Here, we provide a comprehensive breakdown for both Ovis2.5-2B (Lu et al., 2025) and Qwen2.5-VL-3B (Bai et al., 2025).

To rigorously isolate the impact of our task formulation, we introduce two baseline settings:

**Base (Markdown):** Zero-shot inference using standard Markdown output.

**Base (JSON):** Zero-shot inference attempting to follow our structured JSON schema.

As visually summarized for Qwen2.5-VL in Figure 4, and detailed numerically in Table 7 and Table 6, the results reveal a critical insight regarding the difficulty of structured generation:

While the Base (Markdown) models achieve reasonable performance, the Base (JSON) scores drop precipitously. We observed severe schema compliance issues: approximately 50.4% of Ovis2.5 responses and 26.6% of Qwen2.5-VL responses failed to parse as valid JSON. This structural collapse explains the extremely low scores in the JSON baseline.

Our Content-Enriched SFT is crucial not just for domain adaptation, but for enabling the model to master the complex JSON syntax. It eliminates the formatting errors seen in the base model.

The Unified SFT-RL strategy yields the highest performance. By iitializing with the structurally competent SFT model, RL focuses on optimizing holistic rewards.

### B.2. Qualitative Visualization

Figure 6 illustrates the qualitative evolution of the methods on a complex document sample (a dual-column test paper with handwritten answers).

**Baseline Analysis.**   The Qwen-Base model demonstrates basic OCR capabilities, correctly following the global reading order from the left column to the right column. However, it suffers from severe format hallucination. As shown in the left column, it erroneously interprets the entire layout as a rigid table using dense pipe separators "——", which contradicts the document's actual semantic structure. Furthermore, it fails to recognize the high-entropy handwritten content, outputting placeholders like underscores "____" or empty brackets "( )" instead of the student's blue ink answers.

**Unified Improvement.**   The Content-Enriched SFT (SFT-JSON) model successfully corrects the formatting issues. It discards the hallucinatory table syntax and reconstructs the document as a clean, linear sequence of questions. However, consistent with our finding that SFT acts primarily as a "structure learner", it biases towards the printed text and continues to

omit the handwritten answers. For instance, in the poetry completion section, it transcribes the printed questions perfectly but leaves the answer areas blank. The Unified SFT-RL model achieves the best performance by acting as a "content refiner." It retains the correct structure learned during SFT while utilizing the reward signal to actively seek and transcribe the missing semantic details. Crucially, it successfully recovers the handwritten answers that both previous models missed. As seen in the rightmost column, the RL model accurately transcribes the student's filled-in text (e.g., "Bi Jing Xi Hu Liu Yue Zhong..." in the poetry section and "Jì Jìng" in the matching section ), demonstrating high fidelity in both structure and content.

### B.3. Efficiency of the JSON Reformulation.

One potential concern is that the JSON-based formulation may introduce additional structural tokens and therefore increase the auto-regressive decoding cost. To examine this trade-off, we compare the standard SFT-Markdown model with our structure-scaffolded SFT-JSON model on OmniDocBench using Ovis2.5-2B. As shown in Table 8, the JSON reformulation substantially improves the overall parsing accuracy from 56.6 to 82.4, yielding a gain of +25.8 points. Meanwhile, it does not increase the actual inference cost. Instead, the total inference time on OmniDocBench decreases from 25,658 seconds to 21,617 seconds, corresponding to a relative reduction of approximately 15.7%.

Although JSON contains more explicit structural tokens than Markdown, it provides a clearer generation scaffold for the model. In contrast, direct Markdown generation is more prone to formatting collapse and repetition loops, which often cause decoding to continue until the `max_new_tokens` limit. The structured JSON schema helps the model terminate more reliably and reduces such max-token failure cases. Therefore, the JSON reformulation brings a large accuracy gain without increasing the actual decoding cost.

## C. Limitations and Future Works

While our work provides a unified perspective on post-training for document parsing, several limitations remain that open avenues for future research.

**Synthetic Data Gap.**    Our analysis and method rely heavily on a high-quality synthetic data generation pipeline. While this ensures clean ground truth and allows for precise ablation studies, real-world documents often contain noise, artifacts, and handwritten annotations that our synthetic engine may not perfectly replicate. The performance of our "Content-Enriched SFT" on extremely noisy, real-world scans (e.g., faded receipts or historical manuscripts) remains to be fully quantified. Future work should investigate domain adaptation techniques to transfer the structural priors learned from synthetic data to noisy real-world domains.

**Scope of Task.**    Our study is explicitly focused on document parsing—the conversion of visual document images into structured text formats. While parsing is a fundamental prerequisite for many downstream applications, we do not evaluate the model's performance on high-level reasoning tasks, such as Document Visual Question Answering (DocVQA) (Mathew et al., 2021) or Key Information Extraction (KIE) (Jaume et al., 2019), directly within this study. It remains an open question whether the structural and content improvements gained through our unified post-training strategy translate linearly to complex reasoning abilities. Future investigations should expand the evaluation benchmarks to include these downstream tasks to provide a more holistic view of document understanding capabilities.

**Computational Cost of RL.**    Although we employ Group Relative Policy Optimization (GRPO) to mitigate the need for a critic model, the exploration phase of Reinforcement Learning inevitably requires generating multiple samples per prompt. This results in a significantly higher computational cost compared to standard SFT. Developing more sample-efficient RL algorithms or hybrid methods that can achieve the "mode-seeking" behavior of RL with the efficiency of SFT is a critical direction for scaling this approach to larger models and datasets.

**Reward Function Design.**    Our current Structure-Aware RL relies on a weighted sum of Edit Distance, TEDS, and CDM. While effective, these metrics are heuristic proxies for human preference. For instance, Edit Distance may penalize a semantic paraphrase that is factually correct. Future research could explore model-based reward modeling or direct preference optimization (DPO) (Rafailov et al., 2023) to align the model more closely with human semantic understanding of documents, rather than rigid character-level matching.

*Table 4.* Comparison of System Prompts for Markdown vs. JSON Generation. The JSON prompt explicitly enforces structural categorization and specific formatting rules for different element types.

| Task Type | System Prompt Instructions |
| --- | --- |
| **Markdown Generation** | You are an AI assistant specialized in converting PDF images to Markdown format. Please follow these instructions for the conversion:

1. Text Processing: Accurately recognize all text content without guessing. Convert to Markdown while maintaining original structure (headings, paragraphs, lists).

2. Mathematical Formula Processing: Convert all formulas to LaTeX. Enclose inline formulas with `\(` `\)` and block formulas with `\[` `\]`.

3. Table Processing: Convert tables to HTML format. Wrap the entire table with `<table>` and `</table>`.

4. Figure Handling: Ignore figure content; do not describe images.

5. Output Format: Ensure clear structure with appropriate line breaks. Maintain complex layouts as closely as possible.

Please strictly follow these guidelines to ensure accuracy and consistency without adding extra explanations. |
| **JSON Generation** | Please output the layout information from the PDF image, including its category, and the corresponding text content.

1. Layout Categories: The possible categories are `['heading1', 'heading2', 'heading3', 'paragraph', 'table', 'formula', 'image']`.

2. Text Extraction & Formatting Rules:
  • Picture: The text field should be omitted.
  • Formula: Format text as LaTeX.
  • Table: Format text as HTML.
  • All Others: Format text as Markdown.

3. Constraints: Output original text with no translation. All elements must be sorted according to human reading order.

4. Final Output: The entire output must be a single JSON object. |

*Table 5.* Training Hyperparameters for SFT and RL stages.

| Hyperparameter | SFT Stage | RL Stage |
|---|---|---|
| Optimizer | AdamW | |
| Epochs | 1 | |
| Learning Rate | 5e-6 | 5e-7 |
| LR Scheduler | Cosine | Constant |
| Global Batch Size | 256 | |
| Weight Decay | 0.0 | |
| Warmup Ratio | 0.1 | |
| Gradient Accumulation | 16 | |
| Max Sequence Length | 6144 | |
| Precision | bf16 | |
| **RL Specifics** | | |
| Group Size ($G$) | - | 8 |
| KL Coefficient ($\beta$) | - | 0.01 |
| Clip Range ($\epsilon$) | - | 0.2 |

*Table 6.* Performance breakdown of Qwen2.5-VL-3B. The Base (JSON) setting suffers from a 26% format failure rate, resulting in scores far lower than Base (Markdown). The Unified SFT-RL successfully recovers structure and refines content.

| Method | OA ↑ | ED ↓ | CDM ↑ | TEDS ↑ | TEDSS ↑ | Read ↓ |
|---|---|---|---|---|---|---|
| Qwen Base (Markdown) | 72.44 | 0.26 | 79.55 | 63.68 | 26.73 | 0.58 |
| Qwen Base (JSON) | 21.27 | 0.65 | 4.03 | 24.69 | 28.26 | 0.71 |
| Qwen SFT-JSON | 81.09 | 0.13 | 79.16 | 77.20 | 82.19 | 0.15 |
| Qwen SFT-RL | 83.20 | 0.10 | 80.93 | 79.09 | 84.36 | 0.12 |

*Table 7.* Performance breakdown of Ovis2.5-2B. Note that Base (JSON) performs significantly worse than Base (Markdown) due to a high rate of JSON formatting failures ( 50%). Our SFT stage effectively fixes these structural errors.

| Method | OA ↑ | ED ↓ | CDM ↑ | TEDS ↑ | TEDSS ↑ | Read ↓ |
|---|---|---|---|---|---|---|
| Ovis Base (Markdown) | 56.64 | 0.33 | 57.40 | 45.10 | 55.91 | 0.26 |
| Ovis Base (JSON) | 31.71 | 0.54 | 28.05 | 21.18 | 23.01 | 0.47 |
| Ovis SFT-JSON | 82.43 | 0.09 | 80.50 | 75.59 | 81.29 | 0.09 |
| Ovis SFT-RL | 82.53 | 0.09 | 78.62 | 78.07 | 83.83 | 0.10 |

*Table 8.* Efficiency comparison between SFT-Markdown and SFT-JSON on OmniDocBench using Ovis2.5-2B. Total inference time is measured without additional inference acceleration frameworks.

| Method | Overall Accuracy ↑ | Total Inference Time (s) ↓ |
|---|---|---|
| SFT-Markdown | 56.6 | 25,658 |
| SFT-JSON | 82.4 | 21,617 |
| Gain / Reduction | +25.8 | −15.7% |

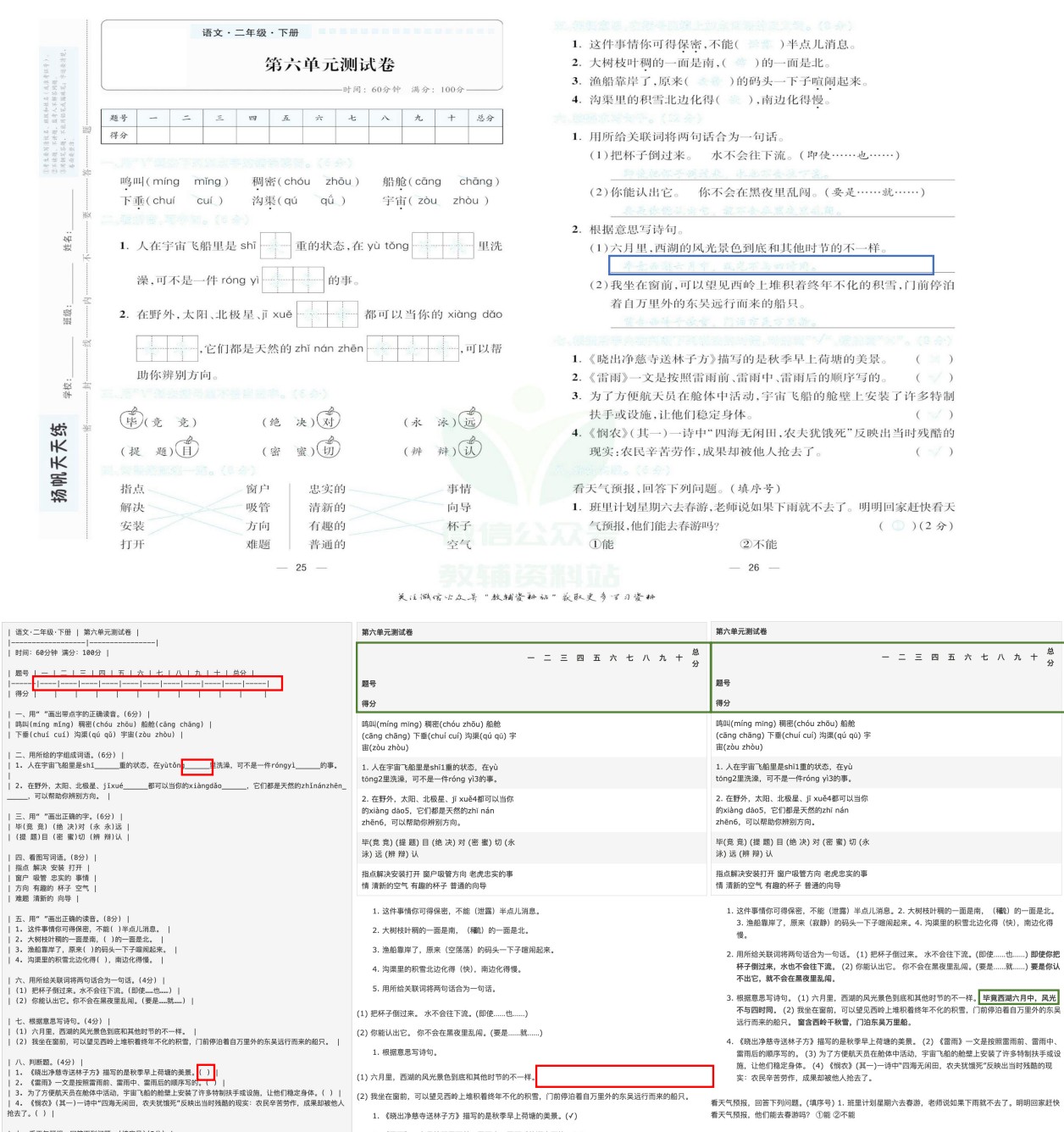

*Figure 6.* Qualitative comparison on a complex test paper with handwritten answers. The top row shows the input image, which features a dual-column layout and blue handwritten student answers. (Left) Qwen-Base suffers from format hallucination, erroneously enforcing a rigid Markdown table structure (indicated by |) that destroys the reading order, and fails to recognize handwritten content. (Middle) SFT acts as a structure learner: it successfully recovers the correct linear layout and removes format hallucinations, yet it treats the document as a blank template, omitting the high-entropy handwritten answers (e.g., leaving the poetry completion blank). (Right) SFT-RL acts as a content refiner: it retains the structural integrity learned during SFT while leveraging the reward signal to accurately transcribe the fine-grained handwritten details (highlighted in green), such as the missing poetry lines "Bi Jing Xi Hu...".

