# OpenReview forum: "Dissecting Post-Training: Uncovering the Complementary Roles of SFT and RL for Document Parsing"
_ICML.cc/2026/Conference — ICML 2026 regular_

### Official Review · Reviewer_5H5U · 2026-03-12

**Soundness:** 3
**Presentation:** 2
**Significance:** 2
**Originality:** 2
**Overall Recommendation:** 4
**Confidence:** 4

**Summary:**

This paper investigates the complementary roles of SFT and RL in document parsing, revealing that SFT primarily acts as a structure learner while RL serves as a content refiner, and proposes a unified framework combining content-enriched SFT with structure-aware RL to improve parsing accuracy.

**Compliance With Llm Reviewing Policy:**

Affirmed.

**Final Justification:**

The authors have addressed my main concerns, especially by providing solid data showing that JSON formatting actually reduces inference time by preventing generation loops.

**Key Questions For Authors:**

- How does the unified SFT-RL pipeline perform on benchmark datasets featuring out-of-domain authentic historical documents compared to the synthetic OmniDocBench?
- Could DPO be integrated to replace the heuristic reward metrics, thereby mitigating the rigid character-level matching bias?
- What is the exact quantitative trade-off between the parsing accuracy gained from JSON formatting and the added auto-regressive decoding time?

**Limitations:**

yes

**Strengths And Weaknesses:**

## Strengths
- The paper provides a rigorous empirical and theoretical analysis using KL-divergence to clearly explain the fundamental inductive bias differences between SFT and RL.
- Transitioning the parsing target from Markdown to JSON format is a clever structural prior that decouples layout prediction from content generation.
- The proposed decomposed reward function helps the model to balance global reading order with local fine-grained element fidelity.

## Weaknesses
- My main concerns lie in the reliance on a purely synthetic dataset for both training and evaluation, which isolates variables but masks the distribution shifts present in real-world scenarios. The JSON structural priors established during SFT might collapse when facing noisy real-world scans with fading, artifacts, or occlusion, leaving the pipeline's true domain adaptation capability untested.
- The heuristic scalar metrics used in the reward function (such as Edit Distance) mechanically penalize completely correct semantic paraphrasing.
- Enforcing verbose and nested JSON templates introduces large token overheads, increasing the model's inference latency and memory costs.

---

> ### Author Rebuttal · Authors · 2026-03-31
>
> We appreciate your recognition of our empirical and theoretical analysis. We will answer the questions below, and we hope this clears up your concerns.
>
> **Q1:** rely on synthetic dataset
>
> **A1:** Please **refer to A1 to Reviewer Ws3i** for more details.
>
> **Q2:** heuristic scalar metrics such as  Edit Distance
>
> **A2:** We thank the reviewer for highlighting this critical distinction regarding evaluation metrics. While this concern is absolutely valid for open-ended generation tasks (e.g., summarization or dialogue), we argue that in the specific domain of **Document Parsing**, Edit Distance and similar heuristic metrics are not just appropriate—they are the **strictly correct objectives**. The fundamental goal of document parsing is **absolute fidelity** to the source image. It acts as an advanced Optical Character Recognition (OCR) system. In this context, "semantic paraphrasing" is actually considered a form of hallucination.
>
> **Q3:**  JSON templates introduces large token overheads
>
> **A3:** We thank the reviewer for raising this important efficiency question. We quantified the trade-off on OmniDocBench using Ovis2.5-2B, comparing the SFT-Markdown model with our structure-scaffolded SFT-JSON model.
>
> Overall parsing accuracy (OA):
> - SFT-Markdown: 56.6
> - SFT-JSON: 82.4
> - Gain: +25.8
>
> Auto-regressive decoding cost (total inference time on OmniDocBench without any framework):
> - SFT-Markdown: 25658s
> - SFT-JSON: 21617s
> - Overhead: +15.7% increase in inference time.
>
> Thus, JSON reformulation brings a large accuracy gain without increasing actual decoding cost. Although JSON introduces more structural tokens, Markdown generation frequently **suffers from formatting collapse and repetition loops**, causing decoding to run until the max_new_tokens limit. In contrast, JSON provides a clear structural scaffold, helping the model terminate reliably and reducing such max-token failure cases.
>
> We will clarify this point in the final version and explicitly report the reduction of repetition-induced decoding overhead.
>
> **Q4:** How does the unified SFT-RL pipeline perform on benchmark datasets featuring out-of-domain authentic historical documents compared to the synthetic OmniDocBench?
>
> **A4:** We thank the reviewer for raising this important generalization question. **As noted in our Limitations and Future Works** (Synthetic Data Gap), evaluating on highly noisy, out-of-domain historical documents remains an open challenge for our current pipeline.
>
> Our study is intentionally scoped to modern/relatively clean documents in order to **isolate the intrinsic learning behaviors of SFT and RL**. Historical documents often contain severe degradation such as faded ink, skew, stamps, and layout damage. Under such conditions, it becomes **difficult to distinguish** whether failures come from the learning objective itself or from upstream visual perception errors. OmniDocBench therefore serves as a cleaner testbed for studying the division of labor between SFT (structure learning) and RL (content refinement).
>
> That said, our framework suggests a clear hypothesis for historical OOD settings: the SFT stage may remain relatively robust in producing valid structural scaffolds, while the RL stage may become more vulnerable to reward noise, since strict character-level rewards could encourage overconfident guesses on illegible content. Extending the pipeline to such domains is an **important next step**, likely requiring stronger degradation-aware augmentation in SFT and uncertainty-aware reward design in RL. We will clarify this more explicitly in the final version.
>
> **Q5:** Could DPO be integrated to replace the heuristic reward metrics, thereby mitigating the rigid character-level matching bias?
>
> **A5:** We thank the reviewer for this insightful question. We agree that DPO is a promising direction, and we have also discussed it in Limitations and Future Works.
>
> In principle, yes—DPO could help mitigate the rigid surface-level bias of heuristic rewards. However, its immediate application to document parsing faces major practical challenges. DPO requires **large-scale high-quality preference pairs**, but document parsing outputs are often extremely long and structurally complex, involving JSON schemas, HTML tables, and formulas. Constructing reliable chosen/rejected pairs for such data is costly and difficult.
>
> By contrast, our current heuristic rewards (Edit Distance, TEDS, CDM) provide dense, automatic, and scalable supervision, which is especially important for exact transcription tasks where every character can matter. Therefore, we view DPO not as a direct replacement, but as a **complementary direction**: heuristic RL is well-suited for learning high-fidelity transcription and structural correctness, while DPO could be introduced later as an additional alignment stage to improve semantic plausibility and readability. We will clarify this trade-off more explicitly in the final version.

---

> > ### Author Rebuttal · Reviewer_5H5U · 2026-04-03
> >
> > The authors have addressed my main concerns, especially by providing solid data showing that JSON formatting actually reduces inference time by preventing generation loops.

---

> > > ### Author Response · Authors · 2026-04-07
> > >
> > > Thank you for your time and consideration. We appreciate your efforts and questions in the review process. We are glad that our response helps fully address your concerns. We promise we will update our paper in the final version. Thanks again.

---

### Official Review · Reviewer_Ws3i · 2026-03-13

**Soundness:** 3
**Presentation:** 3
**Significance:** 3
**Originality:** 3
**Overall Recommendation:** 4
**Confidence:** 4

**Summary:**

The paper examines why SFT often doesn't improve document parsing performance for complex parts like tables and formulas, whereas RL does. The authors point out a fundamental "division of labor": SFT functions as a structure learner, proficient in low-entropy layout syntax, while RL serves as a content refiner, enhancing the fidelity of high-entropy content.

They base this on theoretical analysis, arguing that SFT is a mode-covering forward KL minimization, while RL is a mode-seeking reverse KL minimization. They suggest a two-step plan: Content-Enriched SFT (with JSON format and additional tasks) comes first, followed by Structure-Aware RL (with a broken-down reward function).

This unified approach leads to significant performance improvements on the OmniDocBench benchmark.

**Compliance With Llm Reviewing Policy:**

Affirmed.

**Final Justification:**

I appreciate the authors' engagement across two rounds of discussion.

Because for me the paper's main contribution is an explanation of *why* and *how* SFT and RL behave differently, a large part of its scientific value depends on whether that explanation generalizes beyond the specific setup tested.

I considered lowering my score because the initial rebuttal explained design choices rather than addressing the raised concerns with new evidence, analysis, or clarification. The second round improved on this, so I do not lower.

However, I do not increase my score because the generalization gaps outlined in the discussion mostly remain, and the requested subset analysis was dismissed rather than followed up with a stronger alternative, leaving the mechanistic *how* claim less firmly established.

I therefore still consider it a *Weak Accept*.

**Key Questions For Authors:**

1. Because your RL stage uses character and pixel-level proxies like Edit Distance and CDM, how do you make sure the model is really becoming a "content refiner" instead of just "reward hacking" to fit certain string patterns? Have you conducted any evaluations with human annotators to see whether the RL-refined content is more meaningful than the SFT-JSON outputs?

2. It's been said that base models failed to format JSON correctly up to 50.4% of the time. How much of the "dramatic surge" in performance after SFT-JSON is due to fixing these structural and formatting problems, and how much is due to the model actually getting better at transcribing high-entropy text and formulas?

**Limitations:**

yes

**Strengths And Weaknesses:**

Strengths:

1. The authors tackle the practical problem of document parsing, which is an important first step for many conventional LLM applications.

2. They supported their claims by comprehensive empirical evaluations conducted on two model architectures (but see Weakness 1).

3. They go beyond "data-centric" explanations like scale, quality, and distribution to find that the learning objective/task formulation is the main problem. Their theoretical framework, employing forward versus reverse KL divergence, provides an explanation for the identified behavioral disparities.

4.  The discovery that SFT and RL possess complementary inductive biases is probably applicable to other "structure+content" tasks beyond document parsing.

5. The paper is well-organized, and the figures (like Figures 1 and 2) make it easy to see the "puzzling discrepancy" and how the JSON task reframing changed things

Weaknesses:

1. The evaluation relies heavily on synthetic data, which may not fully capture the noise and artifacts common in scanned documents from the real world.

2. The authors deliberately restrict their experiments to 2B-3B parameter variants, arguing these are common for specialized parsing. As a result, it is unclear whether the "negligible impact of SFT" on direct Markdown still holds in larger models.

3. The paper fails to compare the unified SFT-RL approach with an SFT-only baseline trained with the same amount of compute (e.g., more epochs or larger datasets) to demonstrate that the RL "mode-seeking" behavior is more efficient than brute-force SFT.

4. The SFT-JSON stage fixes formatting mistakes, but the paper says that the base models had failure rates of up to 50.4%. This suggests that the "dramatic surge" in SFT-JSON performance is partly because the baseline did very poorly, not because the content was mastered.

---

> ### Author Rebuttal · Authors · 2026-03-31
>
> We appreciate your recognition of our comprehensive empirical study and writing. We will answer the questions below, and we hope this clears up your concerns.
>
> **Q1:** Rely on synthetic data and model scale
>
> **A1:** We thank the reviewer for the feedback. The goal of our study is to investigate the fundamental learning divergence between SFT and RL. To do this rigorously, our experimental design prioritized controlled variables:
>
> Real-world parsing datasets suffer from severe **annotation noise**, OCR misalignment, and inconsistent formatting. If we used noisy data, it would be impossible to determine whether parsing failures stemmed from data quality or the algorithm itself. Our synthetic pipeline provides a strictly controlled, noise-free environment, with a wide range of CSS stylesheets to isolate algorithm behaviors.
>
> In document parsing, 2B-3B models are the current industry standard, as processing high-resolution images with larger models is **computationally prohibitive**. More importantly, an important concept presented by the manuscript is the theoretical explanation based on KL (Section 4.2). These optimization behaviors are inherently **scale-agnostic** and govern the models regardless of parameter size.
>
> We agree that our conclusions are bound to the document parsing domain. However, it is exactly this domain's unique entanglement of rigid structure and high-entropy content that so clearly exposes the distinct division of labor between SFT and RL. We will explicitly clarify this scope and methodology in the final version.
>
> **Q2:** SFT-RL vs SFT-only with the same amount of compute
>
> **A2:** We agree that this is an important control, and we have already addressed it in Sec. 4.1 under the “Learning with Scaled Data” hypothesis.
>
> Specifically, we include an SFT-Scaled baseline in Figure 2, where the SFT training set is increased from 100k to 300k samples, effectively **giving SFT about 3× more data exposure and training compute**. Despite this, the **gains remain negligible**, especially on high-entropy content such as complex tables and formulas. This shows that RL’s advantage is not simply from more compute, but from a different optimization behavior: SFT’s mode-covering objective still favors low-entropy structural patterns, while RL’s mode-seeking updates better refine exact content. We will make this point clearer in the revision.
>
> **Q3:** “content refiner” rather than reward hacking
>
> **A3:** Document parsing is fundamentally different from open-ended RLHF tasks. It is a deterministic, fidelity-driven problem whose goal is exact reconstruction of the source image. In this setting, rewards such as normalized Edit Distance and CDM are tightly coupled to correctness: a model cannot obtain high reward by producing meaningless but fluent patterns, because **character-level fidelity is itself the target**.
>
> We also explicitly guard against a subtle form of reward imbalance. In preliminary experiments with a single global reward, the model tended to **over-optimize long easy text while neglecting short but difficult formulas**. This is exactly why we introduce Structure-Aware RL with decomposed rewards in Sec. 5.2, so that text, tables, and formulas are optimized separately rather than traded off against each other.
>
> Although we do not include large-scale human evaluation, our qualitative cases provide direct evidence of meaningful refinement. In Figure 6 and A1 to Reviewer d5qN, SFT-JSON omits handwritten content, while RL successfully **recovers the handwritten Chinese** poetry, indicating genuine content improvement rather than reward exploitation. We will add a short discussion on this point in the final version.
>
> **Q4:** How much of the SFT-JSON gain comes from fixing formatting rather than improving content?
>
> **A4:** We agree with the reviewer’s intuition: most of the dramatic gain from SFT-JSON comes from fixing structural formatting failures, rather than from substantially improving high-entropy content transcription.
>
> This is in fact one of our central findings: SFT primarily acts as a structure learner. By learning the low-entropy JSON schema, SFT-JSON sharply reduces format failures and therefore recovers a large amount of performance that the base model loses due to invalid outputs. However, its ability to improve complex content remains limited. As shown in Table 1, SFT-JSON **still struggles on difficult formulas and dense tables**, and only targeted content-enriched sub-tasks bring moderate gains.
>
> Even then, content fidelity plateaus under SFT. As illustrated in Figure 6, SFT can produce the correct structure while still **missing high-entropy handwritten content**. This is exactly where RL is needed: its mode-seeking behavior helps refine precise content beyond the structural scaffold built by SFT. We will clarify in the revision that the SFT-JSON surge is driven mainly by structural recovery.

---

> > ### Author Rebuttal · Reviewer_Ws3i · 2026-03-31
> >
> > **A1.**  The authors explain their experimental choices instead of providing evidence that their findings hold beyond those choices. The strongest move would have been (1) a small real-world validation and (2) even one larger-scale experiment.
> >
> > The "theoretical perspective" does not suffice, as it maps known KL properties onto observed empirical patterns without formalizing intuitions and establishing a causal link. As a minor factual note: while most models in the area are compact, I cannot agree that using models larger than 3B is "computationally prohibitive". Chandra OCR 2 (5B), olmOCR 2 (7B), and others are actively used in this space.
> >
> > **A2.** The weakness referred to a compute-matched comparison. As RL with GRPO generates multiple rollouts per input, computes rewards, and runs policy gradient updates, this is far more than 3x the compute of a single SFT run (likely 15-20x with 8 rollouts). Thus, I am not fully satisfied with this ablation.
> >
> > **A3.** *(It is not my question.)*
> >
> > **A4.** What are the metrics on the subset of outputs where both the base model and the post-trained model produce structurally valid outputs? On this subset, format failures are controlled for, so any remaining gain must reflect genuine content fidelity.

---

> > > ### Author Response · Authors · 2026-04-07
> > >
> > > We sincerely thank you for the prompt follow-up. Your rigorous questions allow us to clarify key details regarding our methodology and data:
> > >
> > > **Regarding A1 (Real-world validation & Model Scale):**
> > >
> > > *   **Real-world Validation:** While our *training* uses synthetic data for strict experimental control and quality, we want to emphasize that our *evaluation* benchmark, **OmniDocBench, is entirely composed of real-world documents**. This directly serves as our real-world validation to prove that our findings hold beyond synthetic scenarios.
> > > *   **Model Scale:** We acknowledge the existence of 5B/7B models. However, the 1B-3B range is deliberately chosen as the most widely adopted "sweet spot" for the performance-cost trade-off in specialized parsing (e.g., **Paddle-OCR-VL 0.9B, MonkeyOCR 3B, DotsOCR 1B, DeepSeek-OCR 3B**). We will explicitly discuss this trade-off and larger models in the Limitations.
> > >
> > > **Regarding A2 (Compute-Matched Baseline):**
> > > We would like to clarify the exact training scale. Our RL stage uses only **~16K samples** (about 1/6 of the base SFT data). Even if we factor in the GRPO rollout multiplier, our SFT-Scaled baseline was trained on **300K samples**. Therefore, the SFT-Scaled baseline actually *exceeds* the total compute of the RL stage. Furthermore, as Figure 2 shows clear performance saturation at 300K SFT samples, simply adding more compute to SFT yields diminishing returns. This confirms that RL’s advantage stems from its mode-seeking algorithmic behavior, not a compute imbalance.
> > >
> > > **Regarding A4 (Subset Evaluation for Content Fidelity):**
> > > We conducted the subset evaluation you brilliantly suggested, but discovered a severe selection bias.
> > > When evaluating solely on structurally valid outputs, the Base model scores artificially high (90.62). This is because the Base model only manages to output valid JSON on the absolutely *easiest* and simplest samples.
> > > In contrast, the post-trained models succeed on a much broader range of layouts, bringing highly complex/hard samples into the "valid subset". Consequently, their average scores on this expanded subset naturally drop (SFT: 86.28, RL: 87.71).
> > > Comparing the Base model to post-trained models here is an apples-to-oranges comparison due to sample difficulty.
> > >
> > > We deeply appreciate your continued engagement, which helps us make these critical nuances clearer in the final manuscript. Thanks again for your questions. We hope our clarification helps address your concerns.

---

### Official Review · Reviewer_d5qN · 2026-03-14

**Soundness:** 3
**Presentation:** 3
**Significance:** 2
**Originality:** 2
**Overall Recommendation:** 4
**Confidence:** 3

**Summary:**

This paper studies post-training for MLLM-based document parsing, focusing on the PDF-to-Markdown setting. The main empirical observation is that standard supervised fine-tuning (SFT) yields only marginal gains over the base model, whereas reinforcement learning (RL) produces substantially larger improvements, especially on challenging elements such as tables and formulas. To understand this gap, the authors run a series of controlled ablations on synthetic image-Markdown pairs and show that adding general-purpose vision-language data, adding more table/formula-heavy samples, or simply scaling the training set does not materially improve SFT. In contrast, reformulating the task from raw Markdown generation to a structured JSON representation leads to a large performance gain. Based on these results, the paper argues that SFT primarily learns low-entropy structural regularities and layout syntax, while RL is better suited for improving high-entropy content fidelity through sequence-level reward optimization. The submission also provides a brief forward-/reverse-KL-based interpretation to motivate this distinction.

Building on this analysis, the paper proposes a unified two-stage pipeline that combines Content-Enriched SFT with Structure-Aware RL. The SFT stage augments JSON-based parsing with auxiliary content-focused tasks such as formula recognition and table transcription, while the RL stage introduces decomposed rewards over text, tables, and formulas to encourage more balanced refinement. The combined SFT-RL approach achieves strong gains over the base model, and the authors present this as evidence that SFT and RL are complementary rather than competing post-training strategies for document parsing.

**Compliance With Llm Reviewing Policy:**

Affirmed.

**Final Justification:**

My overall assessment of this paper is borderline positive. The rebuttal helps clarify the scope and limitations of the work. I do find some of the observations in this paper interesting, particularly the result that SFT on structured intermediate outputs can yield substantial performance gains compared to direct SFT on final prediction. However, the empirical analysis is largely limited to synthetic data, making it difficult to judge how well these findings generalize to real-world settings. In addition, while the theoretical analysis is intuitive and potentially valuable, the empirical verification appears limited, which reduces my confidence in the strength of the submission.

**Key Questions For Authors:**

1. What's the performance comparison between proposed SFT-RL pipeline with a pure RL pipeline with the same rewards? It would be important to show this comparison in the paper.

**Limitations:**

yes

**Strengths And Weaknesses:**

Strength:
1. This paper is well motivated by a clear and interesting empirical puzzle: standard SFT yields only limited gains for document parsing, whereas RL improves performance substantially, especially on harder elements such as tables and formulas.

2. The analysis is reasonably thorough. The authors test several hypotheses for SFT’s weakness, including data scaling, targeted data augmentation, and task reformulation, which makes the empirical study more convincing than a simple comparison of post-training methods.

Weakness:
1. The paper would benefit from more direct qualitative evidence, especially explicit qualitative comparisons between pure SFT and pure RL outputs on the same documents.
2. I understand SFT-RL pipeline improves naively SFT+RL baseline. However, the benefit of the proposed SFT-RL pipeline over a pure RL baseline is not clearly established. There is also no direct comparison between SFT-RL and pure RL baseline.
3. The analysis is limited to relatively small models and only synthetic data, which weakens the generality of the conclusions.
4. The JSON reformulation analysis is somewhat unclear because it changes several aspects of the task at once, and the paper does not discuss how this reformulation would affect RL under a matched setting.
5. The forward-KL / reverse-KL explanation is interesting but not yet strongly validated by the experiments or qualitative examples.

---

> ### Author Rebuttal · Authors · 2026-03-31
>
> We appreciate your recognition of our motivation and empirical study. We will answer the questions below, and we hope this clears up your concerns.
>
> **Q1:** Qualitative evidence and KL explanation.
>
> **A1:** We completely agree that providing direct, side-by-side qualitative comparisons of pure SFT versus pure RL on the exact same document is the most compelling way to substantiate our theoretical claims. To address your point, we selected a complex document containing both hierarchical layouts and dense content. We find that:
>
> - For pure SFT: The model perfectly reconstructs the structural skeleton. For instance, the Markdown with table syntax (<table>) and heading hierarchies are perfectly formatted. However, the SFT model hallucinates or drops specific numbers and uncommon words inside the table cells.
> - For pure RL Output: Conversely, guided by the holistic reward signal, the pure RL model transcribes the exact numbers and text strings with near-perfect fidelity. However, it occasionally misses some LaTeX tags ($$), leading to rendering failures despite the high content accuracy.
>
> This explicit divergence directly shows that SFT tends to "cover the modes" but struggles with diffuse content, whereas RL actively "seeks the mode" of high-reward content correctness but can neglect rigid structural rules. We map the Forward-KL (Mode-covering) and Reverse-KL (Mode-seeking) behaviors directly to our quantitative and qualitative results. SFT guarantees the "average structural safety" (Mode-covering), while RL actively pursues the "peak content fidelity" (Mode-seeking). We will add a dedicated subsection to seamlessly bridge these theoretical insights with the concrete visualizations.
>
> **Q2:** SFT-RL pipeline over a pure RL baseline
>
> **A2:** We acknowledge that while the performance of the pure RL baseline and the SFT-RL pipeline were both reported in the manuscript (in Table 2/Figure 1 and Figure 3/Tables 6 & 7), it is not clear. We have summarized their performance:
>
> |Model|Method|OA|ED|CDM|TEDS|TEDSS|Read|
> |:-|:-|:-|:-|:-|:-|:-|:-|
> |Ovis2.5-2B|PureRL(Decomposed reward)|71.62|0.18|72.20|60.97|69.03|0.16|
> ||OurSFT-RL(JSON)|82.53|0.09|78.62|78.07|83.83|0.10|
> |Qwen2.5-VL|PureRL(Decomposed reward)|84.29|0.09|84.35|77.64|82.92|0.10|
> ||OurSFT-RL(JSON)|83.20|0.10|80.93|79.09|84.36|0.12|
>
> For the Ovis, the SFT-RL pipeline demonstrates a massive improvement over the pure RL baseline across the board. This proves that when a base model lacks strong zero-shot structural priors, pure RL alone is insufficient, and the SFT stage is strictly required to build the structural foundation.
>
> Our empirical analysis (Section 4.1) proved that generating structured JSON elegantly decouples layout from transcription, significantly boosting parsing quality.
> Crucially, **we cannot easily apply pure RL directly to the JSON format**. RL relies on trial-and-error exploration. Because JSON has a highly rigid syntax, pure RL from a cold start suffers from severe formatting failures, resulting in zero rewards and collapsed training. The SFT phase acts as an essential **structural scaffolding**. It provides a warm-started policy that consistently outputs valid JSON syntax, allowing the subsequent RL phase to safely explore and refine the high-entropy content without breaking the structural layout.
>
> We thank the reviewer for this excellent suggestion. We will add a dedicated subsection and a direct comparison table in the revised manuscript.
>
> **Q3:** limited to relatively small models and only synthetic data
>
> **A3:** Please **refer to A1 to Reviewer Ws3i** for more details.
>
> **Q4:** The JSON reformulation analysis.
>
> **A4:** We thanks for the question and clarify the JSON reformulation and its impact on RL as follows:
>
> 1. The transition from Markdown to JSON was designed to **explicitly decouple structure from content**. Markdown implicitly entangles layout syntax (e.g., <tables>) with high-entropy text. The JSON schema simplifies this by forcing the model to explicitly predict a low-entropy "category" (e.g., table, paragraph) before transcribing the "text". This decoupling significantly lowers the structural learning difficulty, allowing SFT to successfully learn the layout.
> 2. If we apply pure RL directly to the JSON task from the base model, **the training suffers from severe reward sparsity and exploration collapse.** Unlike Markdown, JSON syntax is extremely brittle, resulting in a reward of 0 (as shown in A2). In our Appendix Tables 6 and 7, the Base models struggle natively with JSON, exhibiting large format failure rates.
> 3. This "matched setting" failure perfectly illustrates our core argument. Pure RL fails on JSON because it is a poor structure learner; SFT is therefore strictly required as a "structural scaffolder" to reliably fix the JSON schema. Once SFT provides this structurally valid initialization, our Structure-Aware RL can safely act as a "content refiner."
>
> Thanks for the questions. We will add these in the final version.

---

> > ### Author Rebuttal · Reviewer_d5qN · 2026-04-05
> >
> > Sorry for the late reply as my last acknowledgment update is not successful. Most of my concerns are addressed. However, I think my W5 is not fully answered in the authors' responses. I would appreciate it if authors can share some comments. Overall, I maintain positive over the rebuttal and the submission.

---

> > > ### Author Response · Authors · 2026-04-07
> > >
> > > Thank you for your continued engagement and your positive assessment. We apologize that our first rebuttal relied too heavily on a single qualitative example. Here, we map the mathematical properties of KL divergence to the macro-level quantitative metrics observed across our entire dataset:
> > >
> > > 1. Quantitative Validation of Forward-KL (SFT as Mode-Covering):
> > >
> > > - **The Math:** Forward-KL (\mathbb{E}\_{P*} [\log(P*/P_\theta)])  severely penalizes the model if it assigns zero probability to any valid part of the ground-truth distribution. It is structurally forced to "cover all modes."
> > > - **The Dataset-Level Evidence:** This mathematically explains why SFT is exceptionally good at layout generation. The quantitative proof is that **SFT drops the severe format failure rate from ~50.4% (Base) down to near 0%**. It successfully covers all low-entropy structural rules (e.g., JSON schema). Because Forward-KL forces the probability mass to be spread broadly to cover everything, the model lacks the "sharpness" needed for highly specific, complex text. This is quantitatively validated by the fact that **SFT's performance on high-entropy metrics (like CDM for formulas) plateaus significantly**, even when fed more data.
> > >
> > > 2. Quantitative Validation of Reverse-KL (RL as Mode-Seeking):
> > >
> > > - **The Math:** Reverse-KL (\mathbb{E}\_{P_\theta} [\log(P_\theta/P*)]), which underpins RL reward maximization, heavily penalizes generating low-reward samples. It encourages the policy to collapse onto a single, high-reward peak (the specific text string) while actively ignoring secondary modes.
> > > - **The Dataset-Level Evidence:** This theoretically explains why pure RL drives character-level metrics to the extreme. The quantitative proof is the **improvement in CDM and drop in Edit Distance** under pure RL. Because the model "mode-seeks" the content reward, it mathematically ignores non-reward-dominating constraints, leading to structural collapse. This is quantitatively proven by the fact that **Pure RL's structure-sensitive metrics (like TEDS for complex tables) are actually lower than SFT-JSON**, and attempting to run Pure RL directly on the fragile JSON format results in 0 rewards and training collapse.
> > >
> > >
> > > In summary, the macro-level statistical divergence in our tables—where SFT consistently masters structural metrics (Format Error, JSON validity) while RL exclusively breaks through the ceiling of content metrics (CDM, Edit Distance)—is not a random empirical artifact. It is the direct quantitative manifestation of the Mode-Covering vs. Mode-Seeking properties of their respective KL objectives.
> > >
> > > We will explicitly weave this mathematical-to-quantitative mapping in the final version to ensure the KL explanation is rigorously validated by the data, rather than just intuitively described.
> > >
> > > Thank you again for your question and your positive assessment.

---

### Decision · Program_Chairs · 2026-04-30

**Decision:**

Accept (regular)

**Comment:**

This paper analyzes why SFT and RL behave differently in document parsing, showing that SFT learns structure while RL refines content, and proposes a combined SFT→RL pipeline to leverage their complementary strengths. Although reviewers have concerns that the work is limited by reliance on synthetic data and small models, lacks more baseline comparisons, reviewers also agree that the paper is well-motivated, with solid empirical analysis, insightful SFT–RL complementarity, and strong performance gains. Overall, I believe the strengths outweigh the weaknesses, and I lean toward accept.